# Psychiatry Residents’ Attitudes towards Spirituality in Psychiatric Practice in Saudi Arabia

**DOI:** 10.3390/healthcare11233067

**Published:** 2023-11-29

**Authors:** Wid E. Kattan, Aqeel T. Alkhiri, Sultan A. Abughanim, Mohammad T. Taeyb, Maria A. Arafah, Faten N. Alzaben, Maher A. Alhazmi

**Affiliations:** 1Division of Psychiatry, Department of Medicine, College of Medicine, King Abdulaziz University, Jeddah 21589, Saudi Arabia; wakattan@kau.edu.sa (W.E.K.); falzaben@kau.edu.sa (F.N.A.); 2Al-Qunfudah General Hospital, Al-Qunfudah 28821, Saudi Arabia; atalkhiri@moh.gov.sa; 3King Fahad Armed Forces Hospital, Jeddah 23311, Saudi Arabia; 4International Medical Center, Jeddah 23214, Saudi Arabia; mtayb@imc.med.sa; 5Department of Pathology, College of Medicine, King Saud University, Riyadh 12355, Saudi Arabia; 6Makkah Health Cluster, Makkah 24372, Saudi Arabia; alhazmi.m@kamc.med.sa

**Keywords:** education, psychiatry, residents, spirituality

## Abstract

Objectives: This study examined residents’ attitudes and practices regarding the relevance of spirituality in psychiatry within Saudi residency training programs; their experiences and comfort levels in addressing patients’ spiritual concerns; and their interest and past learning experiences in this area of training and practice. Methods: This cross-sectional study targeted trainees and recent graduates of residency programs across Saudi Arabia. The study materials consisted of an electronic questionnaire that was adapted with permission. Results: The total number of respondents was 71 out of 180 potential participants (39.44%). Most residents (64.8%) felt that it was appropriate to inquire about the spiritual aspects of patients’ lives and that it was essential to address spiritual problems or needs that patients may have within the clinical setting (71.8%). Many participants (40.80%) described themselves as being both religious and spiritual. Most respondents (94.4%) did not receive any training on spirituality and psychiatry, and 80.3% said they would like to learn more about the subject. Conclusions: Our findings indicate that residents have an overall high level of personal spirituality and that they feel it is relevant in clinical practice. However, they have not had much training in this area and are interested in learning more. Educational initiatives would be beneficial for improving the effectiveness of residents and patient care in this untapped area of spirituality in psychiatry.

## 1. Introduction

In recent years, spirituality has gained recognition as a clinically relevant aspect of health in general and mental health in particular. This is reflected in the terminology and guidelines referred to in the field of psychiatry. For example, “religious and spiritual problems” are now a diagnostic term available in the *Diagnostic and Statistical Manual for Mental Disorders*, *fifth edition*, under the section of “Problems Related to Other Psychosocial, Personal, and Environmental Circumstances.” [1]. The Association of American Medical Colleges [2] ensures that spirituality is included as part of medical student training, and The Royal College of Physicians and Surgeons of Canada [3] identifies dealing with issues related to culture and spirituality as one of the training objectives in psychiatry that residents must meet.

Many recent studies concerning healthcare professionals’ attitudes towards spirituality have originated in Western countries such as the United States and Canada, but there is a relative paucity of similar studies in the Middle East and Muslim-majority countries [4]. Investigating this matter is important in the case of Saudi Arabia, which is one of the most religious countries worldwide and the birthplace of Islam, but little is known about the dynamics between religiosity and health amongst its people [5]. Thus, studies exploring spirituality and health in Saudi Arabia are considered relevant since it is a society with a rich spiritual and religious culture. Similar to Western countries and in line with its population’s needs, the Saudi Commission for Health Specialties (SCFHS) Saudi Board Psychiatry Curriculum requires that residents must be able to “identify and appropriately respond to relevant clinical issues arising in patient care”, including “culture and spirituality”, and to “establish and maintain clinical knowledge, skills and attitudes appropriate to their practice” [6]. These requirements indicate that residents are expected to effectively address spiritual issues if they are relevant to the patient and to know what is considered an appropriate practice in situations that may be uncomfortable for some.

Although most psychiatrists seem to regard spirituality favorably in terms of its effect on mental health, it was found that only a small number inquired about the spiritual aspects of patients’ lives [7]. Residents are active participants in patient care and have expressed that their perceived incompetence in this area might lead to them ignoring some important spiritual issues [8]. One study showed that residents had a generally positive attitude towards spirituality in psychiatry, despite controversy about some areas, such as boundary violations, ethical considerations, and barriers to discussing spirituality in clinical practice. It also demonstrated that most residents had received little education about the topic but were willing to learn more. These results indicate an educational need that could be addressed by incorporating teaching about spirituality into psychiatry residency training [9].

A more recent study about incorporating religion and spirituality into psychiatry residency training programs in the United Arab Emirates has found that residents who received a six-session course had significantly better scores in pre- and post-test scales for self-reported competence and changes in professional practice. These findings suggest that addressing spirituality in psychiatry residency training programs might improve residents’ attitudes towards spirituality [10].

The aim of this study was to explore psychiatry residents’ attitudes towards spirituality in psychiatry in Saudi Arabia, as well as their current training and training needs in this area. More specifically, the objectives were to assess residents’ attitudes in this interface (whether they were overall positive or negative), their beliefs about how spirituality in psychiatry may influence clinical practice, their experiences as to what degree it comes up in clinical practice, their comfort levels in dealing with this area, their own personal spirituality, their training needs, and how these variables may be related. The results have the potential to inform future research and teaching around integrating spirituality into psychiatry training programs in order to better address the needs of residents and patients.

The results will hopefully help us better understand residents’ perceptions of spirituality, their attitudes towards its relevance in clinical practice and patient care, and their comfort levels in addressing this matter with their patients. The results may inform the efforts of program directors, curriculum planners, and supervisors in developing a learner-centered curriculum on this topic.

## 2. Materials and Methods

***Study design.*** This cross-sectional questionnaire-based study was approved by the institutional review board. The survey was preceded by a consent and a disclosure statement explaining the voluntary and confidential nature of the survey. The survey did not request self-identifying information, email addresses, or IP addresses. Only those who consented proceeded with the survey.

***Participants and sample size.*** The target population of this study was all psychiatry trainees (residents and fellows) enrolled in a training program approved by the SCFHS, as well as recently certified psychiatrists who had completed their training within the past 12 months. The sampling aimed to target the entire psychiatry training body of residents and fellows in Saudi Arabia, which consisted of 180 trainees enrolled in different programs and regions. We anticipated a response rate of 20–50% based on previous studies with similar populations.

***Measures.*** The survey was adapted with permission from a previous study by Kattan and Talwar [9]. The main change was the conversion of questions pertaining to attitude from a 3-point to a 5-point Likert scale to allow the production of an index that can be scored. The questions consisted of Likert-scale items, yes/no questions, “check all that apply” questions, and open-ended items. The survey questions were divided into five sections: (1) demographic information (age, gender, level of training, and regional program), (2) learning experience and interest in future learning, (3) barriers to discussing spirituality and perception of spirituality in terms of their CanMEDS roles, (4) questions on personal spirituality, which were adapted from the intrinsic religiosity subscale of the Duke University Religion Index (IR-DUREL) [11,12], and (5) the Resident Attitude Index (RAI), which looks into resident’s attitudes towards spirituality within their patients in the clinical setting, and which contains questions that tap into four areas: positive attitude, comfort level, experience in practice, and influence on practice. 

IR-DUREL is a 3-item index that uses a 5-point Likert scale. It was used to assess resident’s personal spirituality (not within the clinical setting). The score range is 3–15. The three original items are the following: “In my life, I experience the presence of the Divine (i.e., God)”, “My religious beliefs are what really lie behind my whole approach to life”, and “I try hard to carry my religion over into all other dealings in life.” The index was adapted with permission. The adaptation consisted of removing words (i.e., God) from item 1 and changing the word religious/religion to spiritual/spirituality in items 2 and 3. The survey used, the details of the changes to the original survey, and the scoring manual are available upon request.

***Procedure and data collection.*** The survey was initially piloted among 10 board-certified psychiatrists who finished their training two years before the time of data collection. Their feedback yielded only simple typing errors, which were corrected. The research data were collected using the online survey platform SurveyMonkey^®^. An invitation letter with the survey link was sent to all eligible trainees through their chief residents by email. Two email reminders were sent to participants through their chief residents to improve response rates. Data were collected between April 2019 and October 2019.

***Statistical analysis.*** Data analysis was performed using SPSS 21 (SPSS Inc., Chicago IL, USA). Demographic data were represented by frequencies and percentages to examine sample characteristics. Frequencies were also generated for each response category for individual statements in the questionnaire. The mean RAI was calculated from the respondents’ answers to the 20 statements within the scale. Personal spirituality measured via the adapted IR-DUREL was calculated based on answers to the three questions of this scale. Cronbach’s alpha was generated to examine the reliability of the two indices. Spearman’s rho correlation coefficient was used to test the correlation between the two indices: personal spirituality (IR-DUREL) and the RAI. *t*-tests were used to examine the relationship between these two indices and the training variables (previous training, training level, and interest in training).

## 3. Results

***Characteristics of respondents.*** The total number of respondents was 94 out of 180 residents and fellows in training (52.22%), but 23 respondents’ answers were incomplete and thus excluded. The final analytical set was composed of 71 participants (39.44%). Table 1 outlines the demographic data of the survey respondents.

***Attitudes towards spirituality in practice.*** The Resident Attitude Index (RAI) was found to have good internal consistency (Cronbach’s alpha = 0.76) and was generally favorable towards spirituality in psychiatry (cutoff for positive attitude set at >3, mean = 3.23 ± 0.39, range = 1–5). The five attitudinal subindices based on the conceptual factors are reported in Table 2. Responses to individual items of the RAI can be found in Table 3. Most residents (64.8%) felt that it was appropriate to inquire about the spiritual aspects of patients’ lives and that it was essential to address spiritual problems or needs that patients may have within a psychiatric clinical setting (71.8%). Close to half of the participants (49.3%) felt that both psychiatrists’ and patients’ beliefs might affect the therapeutic relationship. The majority of respondents (73.2%) felt that it was inappropriate to discuss the psychiatrist’s own spirituality with patients when asked, and 31% felt that it was not acceptable to pray with patients if asked, although many were neutral about prayer (45.1%). 

Many participants (38.1%) did not feel comfortable asking patients about their spiritual beliefs, and while others (31.0%) remained neutral, and the remaining (31.0%) felt comfortable with it. However, 47.9% felt that it could be too personal to inquire about such beliefs, and the majority (66.2%) had concerns about the ethical implications of discussing spiritual issues with patients. Many participants (46.4%) reported that spirituality was often cited by patients as a cause of their psychological distress, while most participants said spiritual matters were often mentioned by patients (74.7%) and often cited as related to patients’ ability to cope with psychological distress (71.9%). The majority of participants also reported that spiritual issues were often brought up by patients contemplating suicide (63.4%) and dying patients (57.8%). 

The RAI was not affected by gender, previous training, or training level. However, those with higher attitude scores were more interested in further training, (t(66) = 4.039, *p* ≤ 0.001). Barriers to discussing spirituality in clinical practice and the relation of CanMEDs roles to addressing spirituality are detailed in Table 4.

***Personal spirituality.*** Many participants (40.80%) described themselves as being both religious and spiritual, while 25.40% described themselves as being not spiritual or religious. The rest identified themselves as spiritual only (14.10%) or religious only (8.50%). Personal spirituality measured by means of the IR-DUREL showed good internal consistency (Cronbach’s alpha = 0.83.), and the mean score was 3.63 ± 1.13 (range 3–15). There was a statistically significant positive correlation between personal spirituality measured via the IR-DUREL and the RAI (rs = 0.347; *n* = 63; *p* < 0.005). Personal spirituality did not differ between residents based on gender, previous training, or training level (*p* > 0.05). However, those who were interested in training had higher spirituality scores on the IR-DUREL (t(61) = 2.3; *p* = 0.025)

***Previous training and interest in further training.*** Most respondents (94.4%) did not receive any training on spirituality and psychiatry, and 80.3% said they would like to learn more about the subject. Table 5 shows the types of training residents would be interested in receiving. 

## 4. Discussion

This study aimed to explore residents’ attitudes towards spirituality in psychiatry in Saudi Arabia. The results indicate that attitudes are generally positive, but there remain many areas of uncertainty or controversy. It is interesting to compare this study’s findings with those of other studies that used the same instrument (the original study at a Canadian university [9] and a more recent one conducted at an institute in the US [13]). In terms of personal spirituality, 71% of the Saudi residents reported being religious, spiritual, or both, compared to 62% of Canadian residents and 66.7% of residents in the US survey [9,13]. It seems that across studies, residents differentiate between the concepts of spirituality and religion, and their definitions are debated and numerous in the literature.

Some responses were similar to those in Kattan and Talwar’s study [9]. For example, in both studies, around 71% of residents felt that spirituality was often cited by patients as related to their ability to cope, and around 47% felt that asking patients about spirituality can be too personal or offensive. In both groups, the CanMEDS role of communicator was the most frequently felt to be relevant to addressing patients’ spirituality.

The responses of Saudi residents were closer to those found in the US study in some areas. Saudi and US program residents were more likely to say that patients often mentioned spiritual matters (74.6% and 73.2%, respectively, compared to 31.1% in the Canadian program) and that spiritual matters were brought up by patients contemplating suicide (63.4% and 58.5%, respectively, compared to 33.3% in the Canadian study). On the other hand, the residents in the Saudi program were much more likely than those in North America to report that patients often cited spirituality as contributing to their psychological distress (46.5% compared to 13.3% in the Canadian study and 24.4% in US study). The differences may reflect the different natures of these cultures in terms of spirituality and religiosity, which, in turn, can affect both patient experience of spirituality and resident perception of patients’ spiritual issues. A Gallup study reported that 93% of Saudis considered religion important, compared to 65% of US citizens and 42% of Canadians [14].

Although Saudi residents seem to report that spiritual matters are important to patients, and they are significant in both directions (with positive and negative influences on mental health), they were less likely to feel that it was appropriate to ask about spirituality (64.8% compared to 91.1% of residents in the Canadian program and 87.8% in the US program). They were also less likely to feel comfortable about inquiring into spiritual aspects of patients’ lives (31% compared to 84.4% in the Canadian group and 85.6% in the US group). This could be explained by the finding that the residents in our study were also more likely to have concerns about the ethical implications of discussing spirituality with patients (66.2% compared to 24% in the Canadian group). 

It is interesting to note that Saudi residents seem more likely to perceive patients to be spiritual/have spiritual issues or concerns, which is not surprising given the religious nature of the population, yet they seem less comfortable with bringing up and addressing the issue. Thus, there may be more of a spiritual or “religiosity gap” in Saudi Arabia. This term refers to a discrepancy between patients or clients and mental health professionals, which can interfere with patients being open and seeking help, leading to unmet needs related to recognizing their spiritual issues [15]. Saudi residents were also less likely to believe that a psychiatrist’s personal spirituality can have a bearing on one’s practice when dealing with spiritual issues (40.8% vs. 66.7% among Canadian residents). Some authors would disagree, believing that any interaction between patients and psychiatrists will have a spiritual or religious dimension that could affect the therapeutic relationship.

Differences between the Saudi and North American residents may reflect the training that the latter have received in this area, which may allow them to hold the tension of the paradox: spirituality can have a bearing on one’s practice, but this can be taken into account through awareness and supervision in order to be comfortable addressing the issue. Indeed, Canadian and US residents reported receiving more training (38.6% and 41.75%, respectively) compared to Saudi residents (only 6% reported having any training in this area). In fact, many psychiatry programs in the US and 4 out of 14 responding Canadian programs include mandatory training in religion and spirituality [16]. The importance of training is further highlighted by the finding that when looking at barriers to discussing spirituality, the Saudi residents’ primary barrier was insufficient knowledge/training. In contrast, this was the third most common reason cited by Canadian and US residents, both of whom cited insufficient time as the most common barrier. 

The survey collected quantitative data, and many of the results point to areas of uncertainty and controversy. Interpreting them requires speculation and a deeper understanding. To obtain higher certainty would require qualitative research such as focus groups or in-depth interviews. 

The sample size of this study was small, but this is partly due to the small size of the target population of residents enrolled in Saudi psychiatry programs. The response rate was considered good for an online survey (52.22%), although it dropped to 39.44% after eliminating participants with missing data. It is important to recognize the small sample size as a limitation of the current study. A low response rate is a commonly reported challenge in web-based research surveys. A recent meta-analysis reported an average response rate of approximately 44% for online surveys in education-related fields [17]. In the current study, several factors might have contributed to the low response rate, including the sensitive nature of the topic of spirituality in practice, the length of the survey, and the method of survey distribution. Future studies with larger sample sizes are needed to help draw broader conclusions and enhance the reliability and generalizability of such findings. Potential solutions to address this challenge can include implementing a “mandatory response” feature in electronic surveys, following up with incomplete responses, administering the survey in scheduled teaching sessions or designated time slots, and providing incentives. 

Although the total RAI reliability was good, the Cronbach’s alpha of each subscale was not strong, indicating they are unreliable for further analysis. Thus, reporting relied mostly on the descriptive data of individual items. The sample size was also too small for a meaningful factorial analysis to generate different and more reliable subscales. Nevertheless, the individual items yielded important information, and the subscales were conceptually designed and involved expert opinions from the original study, giving them face validity [9].

To our knowledge, this is the first study on Saudi psychiatry residents looking at spirituality in resident training and residents’ attitudes towards spirituality in psychiatry. One of the strengths of this study was its attempt to modify the questionnaire to obtain an actual attitude index rather than simply reporting on individual items. This was performed by designing responses on a 5-point Likert scale and summing the scores. This attitude index showed good reliability and allowed the correlation of attitude with personal spirituality, which was found to be significant. Thus, our study brings us one step closer to having a validated index to measure residents’ attitudes in this area. Every culture has its nuances and differences, and a replication of this study across more than one culture using a transcultural psychiatry lens might provide a transcultural assessment of attitudes towards spirituality and religion across cultures, and might highlight areas of comparison and differences, in order to help mental health practitioners become more culturally sensitive and better attuned to patient needs.

Interviews and focus groups to gather more nuanced and qualitative data may shed light on areas of controversy and help us understand more about responses that are uncertain or neutral. Also, using the RAI with a larger sample could enable a factorial analysis, proper exploration of its psychometric properties, and determination of whether the conceptual factors are indeed true factors. Future research could also use this index to examine attitudes amongst other groups, such as psychiatrists or residents in other areas that may also have exposure to patients with spiritual concerns, such as oncology, palliative care, and family medicine.

In terms of educational initiatives, there is clearly a desire to learn more about the topic, and there are at least two gaps to address: the gap between what residents experience (if spirituality comes up, it should be addressed) and their practice (they are uncomfortable and have ethical concerns), and the gap between what patients seem to need (addressing spirituality) and what they are receiving (it is not formally addressed). It seems fair to conclude that addressing spirituality in psychiatry training is needed in both didactic and clinical teaching. This was expressed by the residents in this study and aligns well with the training guidelines of the SCFHS and other training and certifying bodies, as well as residents’ expressed wishes.

More information could be gathered through an assessment of formal educational needs in order to design a program to address this need. It is important to reiterate that many programs exist already. They have been described and their benefits have been reported [10,16,18]. Studies also point to these programs being beneficial and well received [19,20]. It should be feasible to build on these initiatives, collaborate with the authors and educators who have implemented them, and adapt them to the culture of Saudi Arabia or any given country and population.

## 5. Conclusions

In conclusion, our findings indicate that residents have a high overall level of personal spirituality and that they feel it is relevant in clinical practice. Yet at the same time, there is some concern and discomfort in actually asking patients about spirituality and addressing patient needs in this area. The residents are interested in learning more and feel it aligns with their communicator role as physicians. Educational initiatives would be beneficial to improve residents’ effectiveness and patient care in this untapped area of spirituality in psychiatry.

## Figures and Tables

**Table 1 healthcare-11-03067-t001:** Demographic characteristics.

Variable	Total (*n* = 71)
*Age*	
Mean ± SD	27.8 ± 2.1
Median (IQR)	28 (26 29)
*Gender*	
Male	32 (45.1%)
Female	39 (54.9%)
*Residency/Fellowship year*	
Resident year 1	14 (19.7%)
Resident year 2	23 (32.4%)
Resident year 3	13 (18.3%)
Resident year 4	11 (15.5%)
Board certified (2018)	9 (12.7%)
Fellows	1 (1.4%)
*Program Region*	
Central region	25 (35.2%)
Western region	29 (40.8%)
Eastern region	17 (23.9%)

IQR: interquartile range, SD: standard deviation.

**Table 2 healthcare-11-03067-t002:** Reliability of the Resident Attitude Index and subindexes.

Scale	Mean	Standard Deviation	Cronbach’s Alpha
Positive attitude	3.74	0.53	0.684
Influence of practice	2.91	0.56	0.467
Comfort level	2.67	0.81	0.690
Experience of practice	3.59	0.51	0.661
Overall attitudes (RAI)	3.23	0.39	0.763
Personal spiritualty	3.63	1.13	0.837

RAI: Resident Attitude Index.

**Table 3 healthcare-11-03067-t003:** Residents’ perceptions about spirituality in psychiatry.

Statements	Strongly Disagree	Disagree	Neutral	Agree	Strongly Agree	Range	Mean	Standard Deviation
	N (%)	N (%)	N (%)	N (%)	N (%)			
*2.1. Positive attitude items*								
As a psychiatrist, it is appropriate to inquire about spiritual aspects of patients’ lives	1 (1.40)	10 (14.10)	14 (19.70)	33 (46.50)	19 (18.30)	1–5	3.66	0.98
As a psychiatrist, it is important to address spiritual problems or needs patients may have within the clinical setting of psychiatry	3 (4.20)	8 (11.30)	11 (15.50)	38 (53.50)	13 (18.30)	1–5	3.64	1.01
Spiritual beliefs can help some patients cope with life stressors	1 (1.40)	2 (7.00)	3 (4.20)	29 (40.80)	36 (50.70)	1–5	4.36	0.81
Spirituality generally has a positive influence on health	2 (2.8)	5 (7.00)	20 (28.20)	27 (38.00)	17 (23.90)	1–5	3.73	0.99
Spiritual beliefs tend to negatively contribute to or compound mental illness by causing guilt, anxiety, or other negative emotions	1 (1.40)	9 (12.70)	30 (42.30)	21 (29.60)	10 (14.10)	1–5	2.57	0.93
Taking patients’ spiritual values into consideration when formulating a treatment plan can improve treatment compliance and success	1 (1.40)	4 (5.60)	11 (15.50)	38 (53.50)	17 (23.90)	1–5	3.92	0.86
As a psychiatrist, it is important to consider the patients’ cultural and community practices when formulating the treatment plan	0 (0.00)	1 (1.40)	5 (7.00)	37 (52.10)	28 (39.40)	2–5	4.29	0.66
*2.2. Influence on Practice Items*								
A psychiatrist’s personal spirituality may have a bearing on his/her practice when it comes to dealing with spiritual issues of patients	6 (8.50)	14 (22.50)	20 (28.20)	22 (31.00)	7 (9.90)	1–5	3.11	1.12
As a psychiatrist, it is acceptable to discuss my own religious beliefs with patients if asked	25 (35.20)	27 (38.00)	13 (18.30)	5 (7.00)	1 (1.40)	1–5	2.01	0.97
As a psychiatrist it is acceptable to pray with patients if asked	9 (12.70)	13 (18.30)	32 (45.10)	15 (21.10)	2 (2.80)	1–5	2.83	0.99
The spiritual beliefs of the psychiatrist and/or the patient may have an effect on the therapeutic relationship	3 (4.20)	16 (22.50)	17 (23.90)	27 (38.00)	8 (11.30)	1–5	3.29	1.07
Depending on patients’ own spiritual orientation, they may perceive the psychiatrist who asks about spirituality as trying to influence patients’ beliefs.	1 (1.40)	9 (12.70)	32 (45.10)	25 (35.20)	4 (5.60)	1–5	3.30	0.82
*2.3. Comfort Level Items*								
I feel comfortable asking patients about the spiritual aspects of their lives	6 (8.50)	21 (29.60)	22 (31.00)	19 (26.80)	3 (4.20)	1–5	2.88	1.03
Asking patients about their spirituality can be too personal or offensive	1 (1.40)	22 (31.00)	14 (19.70)	23 (32.40)	11 (15.50)	1–5	2.70	1.11
I have concerns about possible ethical implications of discussing spiritual issues with patients	1 (1.40)	12 (16.90)	11 (15.50)	39 (54.90)	8 (11.30)	1–5	2.42	0.95
*2.4 Experience of practice Items*								
Depending on patients’ own spiritual orientation, they may perceive the psychiatrist who asks about spirituality	1 (1.40)	9 (12.70)	32 (45.10)	25 (35.20)	4 (5.60)	1–5	3.30	0.82
Spirituality is often cited by patients as related to a cause of their psychological distress	1 (1.40)	10 (14.10)	27 (38.00)	28 (39.40)	5 (7.00)	1–5	3.36	0.86
Spirituality is often cited by patients as related to their ability to cope with psychological distress	0 (0.00)	5 (7.00)	15 (21.10)	43 (60.60)	8 (11.30)	2–5	3.76	0.74
Patients often mention spiritual matters	0 (0.00)	3 (4.20)	15 (21.10)	42 (59.20)	11 (15.50)	2–5	3.85	0.72
Spiritual issues are often brought up by patients contemplating suicide	1 (1.40)	8 (11.30)	17 (23.90)	33 (46.50)	12 (16.90)	1–5	3.66	0.94
Spiritual issues are often brought up by patients who are dying	2 (2.80)	5 (7.00)	23 (32.40)	31 (43.70)	10 (14.10)	1–5	3.59	0.91

N: number.

**Table 4 healthcare-11-03067-t004:** Barriers to discussing spirituality and perceived relevance to CanMEDS roles.

Statement	Number	Percentage
*Barriers to discussing spirituality in clinical practice*		
Insufficient knowledge/training	40	56.3%
Concerns about offending patients	39	54.9%
Insufficient time	29	40.8%
General discomfort	26	36.6%
Concerns about gaining disapproval from other psychiatrists	12	16.9%
*Perceived relevance to CanMEDS roles*		
Communicator	46	64.8%
Professional	38	53.5%
Health advocate	29	40.8%
Medical expert	22	31.0%
Collaborator	18	25.4%
Scholar	14	19.7%
Manager/Leader	10	14.1%

**Table 5 healthcare-11-03067-t005:** Previous training and interest in learning.

Variable	N = (Percentage)
Training in spirituality and psychiatry	
Trained	4 (5.6%)
Not trained	67 (94.4%)
Type of training of spirituality and psychiatry	
Clinical supervision/case-based teaching	2 (2.8%)
Didactic teaching	3 (4.2%)
During interview skills training	0 (0%)
During OSCE training	0 (0%)
In journal club	2 (2.8%)
Others	1 (1.4%)
Spirituality in psychiatry as part of evaluation	
Part of evaluation	9 (12.7%)
Not part of evaluation	59 (83.1%)
Would like to learn more about spirituality in psychiatry	
Yes	57 (80.3%)
No	11 (15.5%)

N: number, OSCE: objective standardized clinical exams.

## Data Availability

Data are contained within the article.

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
