# Peer review of "Psychiatry Residents’ Attitudes towards Spirituality in Psychiatric Practice in Saudi Arabia"

_healthcare, 2023, doi:10.3390/healthcare11233067_

Round 1

Reviewer 1 Report

Comments and Suggestions for Authors

The aim of the study and possible practical outcome should be clearly presented in the beginning of the paper.

In the section “Materials and methods” it was no clearly formulated, if the respondents were asked to present their own attitude towards spirituality or they were asked about the usefulness of including these questions into their interview with patients or

The most interesting part of the study is the section “Discussion”. It seems that transcultural assessment of attitude to spirituality and religion could be the separate objective of the study.

Author Response

Reviewer 2

The aim of the study and possible practical outcome should be clearly presented in the beginning of the paper.

In the section “Materials and methods” it was no clearly formulated, if the respondents were asked to present their own attitude towards spirituality or they were asked about the usefulness of including these questions into their interview with patients or

The most interesting part of the study is the section “Discussion”. It seems that transcultural assessment of attitude to spirituality and religion could be the separate objective of the study.

Responses to reviewer 2:

Thank you for reviewing this paper, and for your clearly outlined comments. We have done our best to respond and take your comments into account. Please find your comments and our responses below:

1.The aim of the study and possible practical outcome should be clearly presented in the beginning of the paper.

The aim and objectives of the study and possible practical outcomes were better clarified towards the end of the introduction as requested. It is provided below for your convenience.

The aim of this study was to explore psychiatry residents’ attitudes towards spirituality in psychiatry in Saudi Arabia, as well as their current training and training needs in this area. More specifically, the objectives were to assess residents’ attitudes in this interface (whether they were overall positive or negative), their beliefs about how spirituality in psychiatry may influence clinical practice, their experiences as to what degree it comes up in clinical practice, their comfort levels dealing with this area, their own personal spirituality, their training needs, and how these variables may be related. The results have the potential to inform future research and teaching around integrating spirituality into psychiatry training programs, in order to better address the needs of residents and patients.

2.In the section “Materials and methods” it was no clearly formulated, if the respondents were asked to present their own attitude towards spirituality or they were asked about the usefulness of including these questions into their interview with patients or

The Materials and methods section was revised and made clearer. Under the “Measures” subheading, we clarified that the Resident Attitude Index (RAI) looks into resident’s attitudes towards spirituality with their patients in the clinical setting, whereas the Intrinsic Religiosity Index of the Duke University Religiosity Index (IR-DUREL) was used to assess resident’s personal spirituality (not within the clinical setting). The paragraph is provided here for your convenience:

The survey questions were divided into five sections:...

5) the Resident Attitude Index (RAI), which looks into resident’s attitudes towards spirituality with their patients in the clinical setting, and which contains questions that tap into five areas: positive attitude, comfort level, experience in practice, and influence on practice.

IR-DUREL is a 3-item index that uses a 5-point Likert scale. It was used to assess resident’s personal spirituality (not within the clinical setting). The score range is 3-15.

3.It seems that transcultural assessment of attitude to spirituality and religion could be the separate objective of the study.

Thank you for this comment. A transcultural assessment of attitude to spirituality and religion would be an interesting objective to explore, and although our study may have tapped into this area, we feel it is an objective beyond what we actually were able to aim for given the relatively homogenous nature of the Saudi culture, and the fact that the study was done within one country. However, your comment inspired us to add the following paragraph to the discussion on page 10.

Every culture will have its nuances and differences, and a replication of this study across more than one culture with a transcultural psychiatry lens might provide a transcultural assessment of attitudes to spirituality and religion across cultures, and may highlight areas of comparison and difference, in order to help mental health practitioners become more culturally sensitive and better attuned to patient needs.

Reviewer 2 Report

Comments and Suggestions for Authors

Thank you for sharing your paper on psychiatry residents' attitudes towards spirituality in psychiatry in Saudi Arabia. I appreciate the effort you put into the research and the clear results and discussions. However, I have a few comments that I would like to share with you. Firstly, in the introduction, it would be better to highlight the aim and objective of the paper more clearly rather than indirectly mentioning it. Secondly, I would like to know what formula was used to calculate the prior sample size. Additionally, the number of actual participants is less than 50% of the total 180 mentioned, which is concerning. Regarding the procedure and data collection, the survey was initially piloted among 10 board-certified psychiatrists who finished their training two years before the time of data collection. It would be helpful to know what changes were implemented based on their feedback. Also, it would be useful to mention the online platform used to collect research data and whether instant messages or emails were used to send invitations to eligible trainees. I also noticed some concerns and discomfort among residents in asking patients about spirituality and addressing patient needs in this area. I wonder if this could be a confounding factor due to the smaller number of respondents. Additionally, I would like to know if there were any reasons for the participants to fill out incomplete surveys and what can be done in the future to limit this problem in future studies. Regarding the questionnaire, the reliability of the Resident Attitude Index and subindexes is not clear as only sub-indices are mentioned. I found it interesting that most respondents (73.2%) felt it was inappropriate to discuss the psychiatrist's spirituality with patients when asked, and 31% felt it was not acceptable to pray with patients if asked, although many were neutral about prayer (45.1%). It would be uplifting and motivating for a patient to believe the doctor because they share the same religion, especially since most people in Saudi follow a common religion. Furthermore, Table 3 is missing the p-value. I believe that considering patients' spiritual values when formulating a treatment plan can improve treatment compliance and success. As a psychiatrist, it is important to consider the patients' cultural and community practices when formulating the treatment plan. Lastly, I found that the following questions were redundant and could be combined: "Overall 2.2 Influence on Practice Items" and "2.4 Experience of practice Items". Also, combining spirituality-related issues such as coping with psychological distress, suicide, and dying could make the paper more concise and less redundant.

Comments on the Quality of English Language

Over all it looks good. Minor grammatical revisions required.

Author Response

Reviewer 3

Thank you for sharing your paper on psychiatry residents' attitudes towards spirituality in psychiatry in Saudi Arabia. I appreciate the effort you put into the research and the clear results and discussions. However, I have a few comments that I would like to share with you.

Responses to reviewer 3:

Thank you for reviewing this paper, and for your clearly outlined comments. We have done our best to respond and take them into account. Please find your comments and our responses below:

1.Firstly, in the introduction, it would be better to highlight the aim and objective of the paper more clearly rather than indirectly mentioning it.

The aim and objectives of the study and possible practical outcomes were better clarified towards the end of the introduction (page2) as requested. It is provided here for your convenience.

The aim of this study was to explore psychiatry residents’ attitudes towards spirituality in psychiatry in Saudi Arabia, as well as their current training and training needs in this area. More specifically, the objectives were to assess residents’ attitudes in this interface (whether they were overall positive or negative), their beliefs about how spirituality in psychiatry may influence clinical practice, their experiences as to what degree it comes up in clinical practice, their comfort levels dealing with this area, their own personal spirituality, their training needs, and how these variables may be related.

2.Secondly, I would like to know what formula was used to calculate the prior sample size. Additionally, the number of actual participants is less than 50% of the total 180 mentioned, which is concerning.

Thank you for your comment. Regarding the first point, our population consisted of residents and fellows training in psychiatry in Saudi Arabia. The entire residents’ body (our study population) at the time of conducting the study consisted of 180 trainees. The survey was sent formally through chief residents of each training center via email to make sure the entire 180 trainees were reached. We received a total of 94 responses, but only 72 were complete enough to include in the final data analysis. We updated the manuscript to further clarify these points. The calculation of a target sample size was not felt to be needed as we were aiming to sample the entire population of interest. This was better clarified in the Materials and Methods section under Participants and sample size on pager 3. Added here for your convenience:

Participants and sample size. The target population of this study was all psychiatry trainees (residents and fellows) enrolled in a training program approved by the SCFHS, as well as recently certified psychiatrists who had completed their training within the past 12 months. The sampling aimed to target the entire psychiatry training body of residents and fellows in Saudi Arabia, which consisted of 180 trainees enrolled in different programs and regions. We anticipated a response rate of 20-50% based on previous studies with similar populations.

Regarding the second point, we understand your concern about the number of actual participants being less than 50% of the total population of 180 residents mentioned in the manuscript. Based on similar previous studies conducted with similar populations, we anticipated a response rate of 20-50%, and indeed got a 39.44% response rate (of complete responses). Additionally, we have added to the discussion (page 11) to address the potential implications of the response rate limitation and addressed its impact on the study findings and generalizability.  Here is the modified paragraph for your convenience:

The sample size of this study was small, but this is partly due to the small size of the target population of residents enrolled in Saudi psychiatry programs. The response rate was considered good for an online survey (52.22%), although it dropped to 39.44% after eliminating participants with missing data. It is important to recognize the small sample size as a limitation of the current study. The low response rate is a commonly reported challenge in web-based research surveys. A recent meta-analysis reported an average response rate of approximately 44% for online surveys in education-related fields [17]. In the current study, several factors might have contributed to the low response rate, including the sensitive nature of the topic of spirituality in practice, the length of the survey, and the method of survey distribution. Future studies with larger sample sizes are needed to help draw broader conclusions and enhance the reliability and generalizability of such findings. Potential solutions to address this challenge can include implementing a “mandatory response” feature in the electronic surveys, following up with incomplete responses, administering the survey into scheduled teaching sessions or designated time slots and providing incentives.

3.Regarding the procedure and data collection, the survey was initially piloted among 10 board-certified psychiatrists who finished their training two years before the time of data collection. It would be helpful to know what changes were implemented based on their feedback.

Thank you for this comment, the piloting feedback only yielded minor typing errors, which were corrected before sending out the final survey version. This point was further clarified in the manuscript under Materials and methods, Procedure and data collection on page 3.

  1. Also, it would be useful to mention the online platform used to collect research data and whether instant messages or emails were used to send invitations to eligible trainees.

Response: Thank you for bringing our attention to this point. We used the online survey platform SurveyMonkey®. As we were not able to obtain the emails of all individual trainees from their programs. Thus we sent the study’s invitation and link through emails to the chief residents and entrusted them to deliver it to the eligible trainees. The chief residents of the programs are usually the point of contact with regards to these matters in training programs in Saudi Arabia. This section was updated to clarify this point under Materials and methods, Procedure and data collection on page 3.

“Procedure and data collection. The survey was initially piloted among 10 board-certified psychiatrists who finished their training two years before the time of data collection. Their feedback yielded only simple typing errors, which were corrected. The research data was collected using the online survey platform SurveyMonkey®. An invitation letter with the survey link was sent to all eligible trainees through their chief residents by email. Two email reminders sent to participants through chief residents to improve response rates. Data was collected between April 2019 and October 2019.”

5.I also noticed some concerns and discomfort among residents in asking patients about spirituality and addressing patient needs in this area. I wonder if this could be a confounding factor due to the smaller number of respondents.

Thank you for your valuable feedback and observation. We agree that the small sample size might indeed be a contributing factor to this finding, and we added to the manuscript that our study's small sample size is a limitation, and emphasized the need for future studies to consider larger sample sizes to draw more accurate conclusions. Having said that, we feel it may actually be a true finding: that many residents have low comfort levels asking patients about spirituality and addressing their spiritual needs. It aligns with our perceived clinical and educational experience here.

In the discussion, we made an effort to discuss the discomfort observed among residents when it came to addressing spirituality with patients. We discussed potential explanations for this discomfort in different paragraphs in the discussion. For instance, we mentioned that Saudi residents might have more concerns about the ethical implications of engaging in discussions about spirituality with their patients. Additionally, we also considered the possibility that the discomfort stems from a lack of training in addressing spiritual needs compared to residents in North America.  See discussion paragraphs 4, 5, 6 page 10.

6.Additionally, I would like to know if there were any reasons for the participants to fill out incomplete surveys and what can be done in the future to limit this problem in future studies.

Thank you for this comment. We updated the manuscript to include some potential reasons for the low response rate or incomplete survey. We also added potential solutions for researchers to consider when conducting future studies. See page 11. Here is the modified paragraph for your convenience:

The sample size of this study was small, but this is partly due to the small size of the target population of residents enrolled in Saudi psychiatry programs. The response rate was considered good for an online survey (52.22%), although it dropped to 39.44% after eliminating participants with missing data. It is important to recognize the small sample size as a limitation of the current study. The low response rate is a commonly reported challenge in web-based research surveys. A recent meta-analysis reported an average response rate of approximately 44% for online surveys in education-related fields [17]. In the current study, several factors might have contributed to the low response rate, including the sensitive nature of the topic of spirituality in practice, the length of the survey, and the method of survey distribution. Future studies with larger sample sizes are needed to help draw broader conclusions and enhance the reliability and generalizability of such findings. Potential solutions to address this challenge can include implementing a “mandatory response” feature in the electronic surveys, following up with incomplete responses, administering the survey into scheduled teaching sessions or designated time slots and providing incentives.

7.Regarding the questionnaire, the reliability of the Resident Attitude Index and subindexes is not clear as only sub-indices are mentioned.

Thank you for this comment. The row for the “overall attitudes (RAI) in table 2, page 5 addresses reports on reliability. We also described this result in the text to further clarify (page 5), and defined the abbreviation to make it clear. Here is the paragraph for your convenience:

Attitudes towards spirituality in practice. The Resident Attitude Index (RAI) was found to have good internal consistency (Cronbach’s alpha = 0.76) and was generally favorable toward spirituality in psychiatry (cutoff for positive attitude set at >3, mean = 3.23 ± 0.39, range = 1-5).

8.I found it interesting that most respondents (73.2%) felt it was inappropriate to discuss the psychiatrist's spirituality with patients when asked, and 31% felt it was not acceptable to pray with patients if asked, although many were neutral about prayer (45.1%). It would be uplifting and motivating for a patient to believe the doctor because they share the same religion, especially since most people in Saudi follow a common religion.

Thank you for this comment. We agree that these findings are interesting; that in a population where the doctors and patients share the same religion, residents still feel it’s inappropriate to discuss spiritual issues. The paper points to some of the barriers to discussing spirituality in Table 4 (page 5), and points in the discussion to the possibility of lack of training as an explanation as well. More in-depth focus groups and qualitative studies could give us a deeper understanding of the issue. We also agree that, as a lot of research has shown, addressing spiritual issues in a professional ethical manner can be uplifting and beneficial to patients.

  1. Furthermore, Table 3 is missing the p-value.

Thank you for your comment regarding Table 3 in our study. We appreciate your attention to detail.  Table 3 shows the relevant statistics for the item scores for the Positive attitude, Influence on Practice, Comfort Level, and Experience of practice scales. Table 3 reports descriptive statistics; the counts, percentages, means, and standard deviations for these items associated with scales mentioned earlier. It was deemed that a P-value was not required to be reported in Table 3. We would be happy to perform any needed analysis.

  1. I believe that considering patients' spiritual values when formulating a treatment plan can improve treatment compliance and success. As a psychiatrist, it is important to consider the patients' cultural and community practices when formulating the treatment plan.

We agree, and this is in line with the training and practice guidelines in the US, Canada and Saudi Arabia, as mentioned in the introduction. We believe (and there is research to show as well) that there are gaps between patients’ needs and what is being provided, and between what the guidelines say and what the training programs are delivering.

  1. Lastly, I found that the following questions were redundant and could be combined: "Overall 2.2 Influence on Practice Items" and "2.4 Experience of practice Items". Also, combining spirituality-related issues such as coping with psychological distress, suicide, and dying could make the paper more concise and less redundant.

We agree that the items in 2.2 and 2.4 might be combined and perhaps shortened, maybe under a broader heading of spirituality in practice or spirituality in the clinical setting. Spiritual coping and psychological distress can also be combined under one concept that looks at the role of spirituality in both stress and coping. It would be worth looking into revising the questionnaire in future studies to make it more succinct and concise.

Reviewer 3 Report

Comments and Suggestions for Authors

Feedback on Healthcare 2689244, 17 November 2023

This presentation was well organized. There was a reasonable number of participants which makes the paper worth reading (have some face validity). The text was well organized and discussion flowed well. Overall, the paper presents engaging script laying a relatively sound basis and procedure for more extensive subsequent studies and practices to be implemented.

With adequate attention to the following minor points, this paper would be considered satisfactory for publication, in this reviewer’s opinion:

Page 7, description of item in 2.4 is incomplete’ … they may perceive the psychiatrist who asks about spirituality (as what?)’

Page 8 , Table 5 There appears to be an inconsistency between the first category, which indicates 4 ‘trained’, and the third category, ‘impact of training’ which has 6 responses. Please explain the apparent increase in number of responses/respondents.

Author Response

Reviewer 4

This presentation was well organized. There was a reasonable number of participants which makes the paper worth reading (have some face validity). The text was well organized and discussion flowed well. Overall, the paper presents engaging script laying a relatively sound basis and procedure for more extensive subsequent studies and practices to be implemented.

With adequate attention to the following minor points, this paper would be considered satisfactory for publication, in this reviewer’s opinion:

Page 7, description of item in 2.4 is incomplete’ … they may perceive the psychiatrist who asks about spirituality (as what?)’

Page 8 , Table 5 There appears to be an inconsistency between the first category, which indicates 4 ‘trained’, and the third category, ‘impact of training’ which has 6 responses. Please explain the apparent increase in number of responses/respondents.

Response to reviewer 4:

Thank you for your constructive feedback and clearly outlined comments. We have done our best to address them in order to improve the paper.

1.Page 7, description of item in 2.4 is incomplete’ … they may perceive the psychiatrist who asks about spirituality (as what?)’

Thank you for pointing this omission out. The sentence was completed on page 7 and page 16 where it was also cut off by the table.

The full sentence added is:

Depending on patients' own spiritual orientation, they may perceive the psychiatrist who asks about spirituality as trying to influence patients' beliefs.

2.Page 8 , Table 5 There appears to be an inconsistency between the first category, which indicates 4 ‘trained’, and the third category, ‘impact of training’ which has 6 responses. Please explain the apparent increase in the number of responses/respondents.

Thank you for pointing out this error in the numbers. Based on our careful examination of the data, we discovered that a few participants may have responded inaccurately to a specific question. We would like to note that the number of participants who received training and were expected to answer this question was extremely small (4). As a result, we made the decision to exclude this particular question from the table. We have discussed this with the statistician and are assured that this removal will not have any significant impact on the reported results or the discussion sections.
